# Generating Data for Symbolic Language
# with Large Language Models

**Jiacheng Ye♠, Chengzu Li♣, Lingpeng Kong♠, Tao Yu♠**
♠The University of Hong Kong    ♣University of Cambridge
{jcye2,lpk,tyu}@cs.hku.hk, cl917@cam.ac.uk

## Abstract

While large language models (LLMs) bring not only performance but also complexity, recent work has started to turn LLMs into data generators rather than task inferencers, where another affordable task model is trained for efficient deployment and inference. However, such an approach has primarily been applied to natural language tasks, and has not yet been explored for symbolic language tasks with complex structured outputs (e.g., semantic parsing and code generation). In this paper, we propose SYMGEN which utilizes LLMs for generating various annotation-expensive symbolic language data. SYMGEN consists of an informative prompt to steer generation and an agreement-based verifier to improve data correctness. We conduct extensive experiments on six symbolic language tasks across various settings. Compared with the LLMs, we demonstrate the 1%-sized task model can achieve comparable or better performance, largely cutting inference and deployment costs. We also show that generated data with only a few human demonstrations can be as effective as over 10 times the amount of human-annotated data when training the task model, saving a considerable amount of annotation effort. SYMGEN takes a step toward data generation for annotation-expensive complex tasks, and we release the code at https://github.com/HKUNLP/SymGen.

## 1 Introduction

In the natural language processing (NLP) literature, the march of scaling language models has been an unending yet predictable trend, with new models constantly surpassing previous ones in not only performance but also complexity (Radford et al., 2019; Brown et al., 2020; Chowdhery et al., 2022). Such large language models (LLMs), however, incur a large computational cost in practice, especially when deployed in resource-restricted systems and inference in low-latency applications (Bommasani

Figure 1: Sample symbolic language datasets with complex structured outputs. The names of the symbolic languages are shown in square brackets.

et al., 2021).

Instead of treating LLMs as edge task inferencers, a recent line of work leverage LLMs as data generators, with the generated data being used to train more affordable task-specific models for efficient deployment and inference (Schick and Schütze, 2021; Meng et al., 2022; Ye et al., 2022b, *inter alia*). With only a few or even without demonstration examples, the LLMs can generate high-quality data via in-context learning (Brown et al., 2020) or prompting (Radford et al., 2019). The task models trained on these generated data can achieve comparable or even better performance than the LLMs and enjoy a low inference cost at the same time.

However, previous work mainly focuses on generating natural language data. To what extent this approach works for complex structured data, such as meaning representation and codes (Figure 1), remains an open question. The investigation of data generation via LLMs in the context of such symbolic language tasks is also extremely intrigu-

ing for two reasons: 1) the human annotation procedure for these tasks requires expensive domain expert efforts (Clarke et al., 2010) and carefully-designed strategies (Wang et al., 2015; Iyer et al., 2017; Herzig and Berant, 2019, *inter alia*); 2) conventional data augmentation methods aiming at enriching datasets for these tasks require handcrafted rules, a considerable number of expert demonstration examples, and are mostly task-specific (Jia and Liang, 2016; Yu et al., 2018a; Andreas, 2020, *inter alia*).

To address these issues, we propose **Sym**bolic data **Gen**eration (SYMGEN) for various annotation-expensive symbolic language tasks. SYMGEN works with an LLM trained on code (i.e., Codex-175B; Chen et al. 2021) and optional task-specific structure knowledge (e.g., database for SQL; Iyer et al. 2017) through prompting or in-context learning. SYMGEN also comprises an agreement-based verification module, in which the outputs are verified by execution (e.g., programs, logical forms) or formatting (e.g., pseudo-logical forms), to ensure high-quality generations. With the generated data, we train efficient task models with around 1% size of Codex for task inference (e.g., T5 with size 770M and 3B; Raffel et al. 2020).

We experiment on six symbolic languages which are SQL, Bash, Python, Prolog, Task Oriented Parsing representation (TOP-representation; Gupta et al. 2018) and Question Decomposition Meaning Representation (QDMR; Wolfson et al. 2020). We consider generating data in zero-shot, few-shot, and full data settings. Our key research findings include the following:

- Compared with the gigantic LLM inferencer, the approximately 1%-sized task inferencer can achieve comparable or superior performance, producing state-of-the-art performance in some tasks (§3.4);

- Compared with human annotations, the performance on few-shot generated data is comparable with that on more than 10x or 100x human-annotated data, greatly reducing annotation effort for complex tasks (§3.5);

- In a zero-shot setting, the small task model can surpass the same model trained with full human annotations as well as the larger Codex on specific symbolic language such as SQL (§3.6);

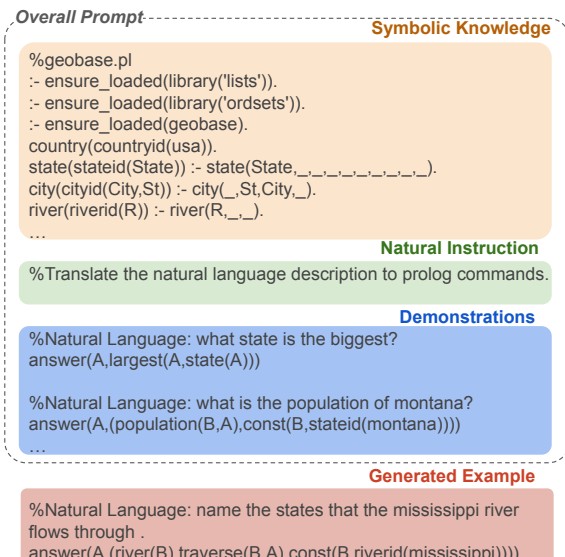

Figure 2: An example of an overall prompt that consists of symbolic knowledge, natural instruction and demonstrations, and a newly generated example by Codex for the GeoQuery dataset.

- During data generation for symbolic language, symbolic knowledge and demonstrations have a greater impact than natural language instructions, and verification is essential to ensure data quality (§3.7, §4.2).

## 2 SYMGEN

To automatically curate numerous data for various annotation-expensive symbolic language tasks, we propose a unified pipeline, named SYMGEN. SYMGEN comprises data generation by prompting LLMs and data verification by executing or formatting.

### 2.1 Prompt-based Generation

Human annotators need to carefully review the annotation instructions to perform annotation, and the same is true for LLMs. We include natural language instructions, task-related symbolic knowledge (i.e., database, ontology), and a few labeled examples into prompt construction to steer the generation. An example of the prompt is shown in Figure 2. We display prompts for each task in Appendix G.

Different from previous work in classification (Schick and Schütze, 2021; Meng et al., 2022; Ye et al., 2022b) where one of the limited label descriptions is used to guide the generation of $\mathbf{x}_i$ for classification tasks, the output structures for symbolic language is unenumerable and requires task-

specific strategies to construct. Hence, we first generate input $\mathbf{x}_i$ and then the output $\mathbf{y}_i$ conditioned on the generated $\mathbf{x}_i$. LLMs may generate erroneous outputs due to not satisfying different grammatical constraints defined in different symbolic languages. Therefore, we over-generate multiple candidates for further verification. After prompt-based generation, we have a dataset $\mathcal{D} = \{(\mathbf{x}_i, \{\mathbf{y}_{i,j}\})\}$ for each task.

## 2.2 Agreement-based Verification

In this work, we adopt an over-generation and verification approach to improve generation quality. Formally, given a set of sampled output answers $\{\mathbf{y}_{i,j}\}$ for input $\mathbf{x}_i$, we verify each answer $\mathbf{y}_{i,j}$ by calculating:

$$w_{i,j} = \sum_k \text{sim}(\text{exec}(\mathbf{y}_{i,j}), \text{exec}(\mathbf{y}_{i,k})), \quad (1)$$

where $\text{exec} = (\cdot)$ is a task-specific execution or formatting function (e.g., executing python program, formatting QDMR into graph representation), $\text{sim} = (\cdot, \cdot) \in [0, 1]$ is a similarity function to compare the two results after running function exec. A large value of $w_{i,j}$ indicates the $j$-th answer is highly similar to others, and is thus less prone to be mistakenly labeled. The value of the most confident answer $w_i = \max w_{i,j}$ is used to measure the quality of the input-output pair, and we only keep those with $w_i$ larger than a certain threshold $T$, indicating the input-output pair is sufficiently confident.

In practice, when performing exec, we discard $\mathbf{y}_{i,j}$ that fails in exec, which means it contains grammatical errors. When using Exact-Match (EM) as the similarity function, the similarity score ranges in $\{0, 1\}$, with 1 indicating that the two execution results are exactly the same. If multiple answers have the same value, we choose the answer that has the maximum log-likelihood during generation.

## 3 Experiments

### 3.1 Datasets and Evaluation Metrics

We consider five datasets that cover a range of programming languages and symbolic meaning representations: Spider (SQL; Yu et al. 2018b), NL2Bash (Bash; Lin et al. 2018), MBPP (Python; Austin et al. 2021), MTOP (TOP-representation; Li et al. 2021) and Break (QDMR; Wolfson et al. 2020). We summarize the choice of the execution

| Dataset | exec(·) | sim(·,·) | Evaluation |
|---|---|---|---|
| Spider | Execution | EM | EM, EX |
| NL2Bash | Bashlex | BLEU | BLEU |
| MBPP | Execution | EM | EX |
| GeoQuery | Execution | EM | EM, EX |
| MTOP | TOP Tree | EM | EM, Template |
| Break | QDMR Graph | EM | LF-EM |

Table 1: Summary of evaluation metric(s), execution function $\text{exec}(\cdot)$ and similarity function $\text{sim}(\cdot, \cdot)$ used in verification module for each task. EM and EX refers to Exact-Match and Execution accuracy.

or execution function exec, similarity function sim, and evaluation metrics for each dataset in Table 1. Details of the datasets and evaluation metrics are illustrated in Appendix A.

### 3.2 Comparison Methods

We generate data under various settings such as zero-shot and few-shot, and then train task models, e.g., T5-large and T5-3B, for inference. We compare the performance of the task models with both LLM inferencers and the task models that are directly finetuned with human-annotated data rather than LLM-generated data:

- **Codex** (Chen et al., 2021). The tuning-free method that performs prompt-based in-context learning with Codex. Due to the length restriction, we perform prompt retrieval to include as many similar examples as possible in the full data setting.
- **Codex + Verification**. The method is similar to the above but further includes the answer verification module as discussed in § 2.2.
- **T5-Large** (Raffel et al., 2020). The tuning-based method that directly fine-tunes T5-large model with few or full human-annotated data instead of generated data.
- **T5-3B** (Raffel et al., 2020). The same method as the one above, but using a T5-3B model.
- **SOTA**. The state-of-the-art models for each dataset. The models and the corresponding number of parameters are Spider (Scholak et al. 2021; 3B), NL2Bash (Shi et al. 2022; 175B), MBPP (Chen et al. 2022; 175B), MTOP (Xie et al. 2022; 3B), Break (Hasson and Berant 2021; ∼300M) and GeoQuery (Qiu et al. 2022b; 11B).

| Model | Spider | | NL2Bash | MBPP | GeoQuery | | MTOP | | Break | Δ |
|---|---|---|---|---|---|---|---|---|---|---|
| | EM | EX | Char-BLEU | EX | EM | EX | Templete | EM | LF-EM | |
| *Full data setting* | | | | | | | | | | |
| #Human annotations | 7,000 | | 8,090 | 374 | 600 | | 15,667 | | 44,321 | |
| #SYMGEN | 75,845 | | 47,803 | 36,367 | 44,266 | | 36,085 | | 46,839 | |
| SOTA | **75.50** | 71.90 | 58.50 | **67.90** | **93.60** | - | 87.74 | 83.76 | 46.90 | |
| Codex | 58.03 | 64.41 | 67.68 | 60.00 | 79.64 | 94.29 | 85.77 | 78.61 | 47.40 | |
| Codex + Verification | 60.54 | 70.21 | **75.40** | 65.56 | 79.29 | 94.64 | 87.20 | 80.63 | 50.10 | ↑3.08 |
| T5-Large | 66.63 | 64.12 | 65.95 | 13.33 | 83.93 | 91.79 | 86.85 | 83.04 | 52.70 | |
| T5-Large + SYMGEN | 70.21 | 69.63 | 67.17 | 43.33 | 84.64 | 96.07 | **88.59** | **84.83** | 54.50 | ↑5.68 |
| T5-3B | 71.76 | 68.38 | 65.97 | - | 83.21 | 89.29 | 87.74 | 83.76 | 53.30 | |
| T5-3B + SYMGEN | 73.40 | **73.11** | 67.26 | - | 84.29 | 96.07 | 88.50 | 84.47 | **55.20** | ↑2.36 |
| *Few-shot setting* | | | | | | | | | | |
| #Human annotations | 10 | | 10 | 10 | 10 | | 10 | | 10 | |
| #SYMGEN | 77,818 | | 39,585 | 46,793 | 31,968 | | 40,673 | | 41,385 | |
| Codex | 53.77 | 65.76 | 61.58 | 58.89 | 39.64 | 65.00 | 27.52 | 18.97 | 26.10 | |
| Codex + Verification | 53.97 | **67.89** | **64.16** | 67.78 | 41.79 | 72.50 | 29.31 | 20.49 | 28.20 | ↑3.21 |
| T5-Large | 0.00 | 0.00 | 17.65 | 0.00 | 10.00 | 12.86 | 8.01 | 4.25 | 0.60 | |
| T5-Large + SYMGEN | 51.84 | 59.48 | 57.83 | 35.56 | 43.21 | 77.14 | 30.34 | **23.85** | 30.30 | ↑39.58 |
| T5-3B | 0.97 | 1.26 | 28.11 | - | 7.86 | 10.71 | 5.50 | 2.50 | 1.20 | |
| T5-3B + SYMGEN | **58.51** | 66.83 | 57.21 | - | **44.64** | **77.50** | **30.56** | 23.49 | **31.10** | ↑41.47 |

Table 2: Results of data generation for training a task model under full data and few-shot settings. The top-scored results for each setting are **bold**. We show the average improvement with SYMGEN across all tasks in the last column.

## 3.3 Implementation Details

We use code-davinci-002 version for Codex and T5-large and T5-3B for task models. Details are elaborated in Appendix B.

## 3.4 SYMGEN + T5 vs. Codex Inferencer

In this section, we consider generating data in few-shot and full data settings and report the model performance in Table 2. Firstly, we can see the performance of T5-Large consistently increases after adding data from SYMGEN on all tasks. Notably, we can achieve an on-average 40% performance boost in the few-shot setting. Secondly, though prompting-based inference has become the de-facto standard to use LLMs on downstream tasks, we find use LLMs as data generators and training a much smaller task model can achieve comparable (e.g., Spider and NL2Bash) or better (e.g., Geoquery, MTOP, and Break) performance. The reasons can be twofold: 1) As recent work proves in-context learning is an extreme approximation of fine-tuning with a single-step gradient descent (von Oswald et al., 2022; Dai et al., 2022), LLM inferencer fails in utilizing the valuable human annotations, even with prompt retrieval. For example, Codex can surpass T5-3B on Spider in a few-shot setting but cannot in full data setting; 2) the obtained knowledge (i.e., generated data) from interacting with the verifier is not explicitly learned by the LLMs, mean-

ing it never learns to correct its own mistakes, and such knowledge also improves LLMs themselves as shown in Haluptzok et al. (2022). In comparison, the task model can learn from those successful interactions. Finally, we find an exception on MBPP, where Codex inferencer significantly outperforms T5, indicating that long-code generation is still challenging for small-sized models.

## 3.5 SYMGEN vs. Human Annotations

A key benefit of SYMGEN is reducing annotation effort when training a task-specific model. We show the performance of the trained T5-large model under various scales of human-annotated and few-shot generated data by SYMGEN in Figure 3. When using human annotations, the model performance grows linearly with exponentially increased data size, which mirrors the power-law in neural models (Kaplan et al., 2020). While for 10-shot generated data, the slope, which indicates the quality of generated data, varies for different symbolic languages. For example, it's relatively easier for Codex to generate SQL and Bash than TOP-representation and QDMR, which we hypothesize is due to the large amount of SQL and Bash commands in the pretraining GitHub corpus (Chen et al., 2021). In the extremely low resource scenario where only 10 human annotations are given, the performance SYMGEN can achieve, as indicated by

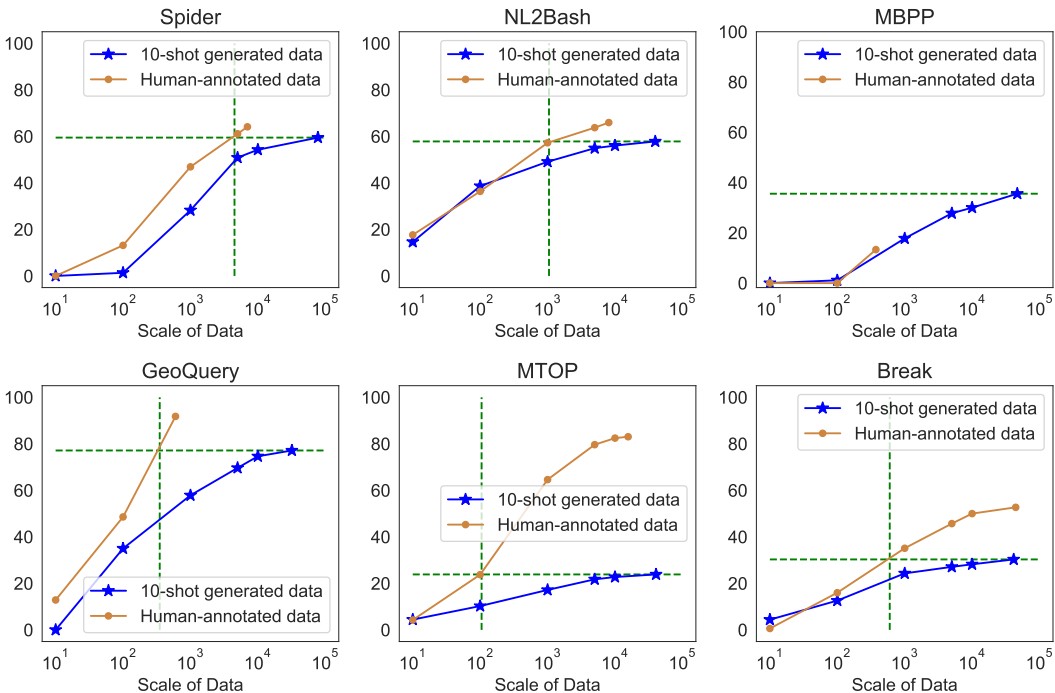

Figure 3: T5-large performance trained under various scales of human-annotated and few-shot generated data by SYMGEN. The horizontal line indicates the performance of the model trained on SYMGEN with only 10 initial human-annotated examples, which is comparable with that on more than 10x (GeoQuery, MTOP, Break) or 100x (Spider, NL2Bash, MBPP) human annotations.

the green horizontal line, significantly outperforms the model trained solely on these 10 given data points. Moreover, the intersection point of the horizontal and vertical lines indicates the performance achieved by training the model on the data generated by SYMGEN is comparable to that on at least 100 (e.g., MTOP) and up to several thousand (e.g., Spider) human-labeled data. This shows the potential of SYMGEN to greatly reduce the annotation effort on complex tasks.

### 3.6 SYMGEN for Zero-shot Learning

Given the striking ability of SYMGEN in few-shot data generation, we take a step forward to see whether it can generate high-quality dataset without any human annotations. We found it hard to control the format in generating most symbolic languages without demonstrations, but we succeed in generating SQL, as shown in Table 3. We find with appropriate prompt and verification, one can achieve a high zero-shot performance of 67.21, outperforming the supervised T5-Large model. We also note that the EM metric is much lower than that of the T5 models, indicating Codex mostly generates grammatically different but semantically correct SQLs. Note for pre-trained models, leakage of the test data is a potential concern (Barbalau

| Model | EM | EX |
|---|---|---|
| *Full data setting* | | |
| T5-Large | 66.63 | 64.12 |
| T5-3B | 71.76 | 68.38 |
| SOTA (Scholak et al., 2021) | 75.50 | 71.90 |
| *Zero-shot setting* | | |
| #Human annotations | | 0 |
| Codex | 45.45 | 64.89 |
| Codex + verification | 45.65 | 67.21 |
| T5-Large | 0.00 | 0.00 |
| T5-Large + SYMGEN (140db, 71k) | 45.74 | 56.38 |
| T5-Large + SYMGEN (160db, 103k) | 50.29 | 65.67 |
| T5-3B | 0.00 | 0.00 |
| T5-3B + SYMGEN (140db, 71k) | 48.55 | 61.03 |
| T5-3B+ SYMGEN (160db, 103k) | **53.38** | **69.25** |

Table 3: Results for zero-shot data generation on Spider. We generate 71k data using databases from the training set (140 databases), and 103k data using both the training and development sets (a total of 160 databases).

et al., 2020; Carlini et al., 2021; Rajkumar et al., 2022). Based on the much lower EM accuracy, we attribute the success of zero-shot learning to prompt engineering rather than memorization.

Moreover, it has been shown that adapting to the new environment significantly outperforms data augmentation in the training environment by Zhong et al. (2020b). Given no human-annotated data on

the development environment, we further generate data for those 20 databases as surrogate knowledge for adaptation. We can see the results significantly increase after training on those additional data, and even outperform the large Codex as well as the human-supervised T5-Large model, indicating SYMGEN can be used for zero-shot adaptation for specific symbolic languages such as SQL.

### 3.7 Prompt Engineering in SYMGEN

Recent work highlights PLM sensitivity to the natural instructions (Zhao et al., 2021; Liu et al., 2022; Gao et al., 2021). In this section, we study the influence of symbolic knowledge (e.g., database and ontology), natural instruction, demonstration, and language reformulation on answer generation. An example of these four types of information in prompts is shown in Figure 2. We report the results of removing certain types of information in Table 4. Removing symbolic knowledge or demonstrations has a greater impact on the answer quality than natural instructions, suggesting symbolic language prediction benefits more from the provided symbolic knowledge and exemplar pairs. An exception is on Spider where removing demonstrations slightly hurt performance, which is mainly because Spider is a cross-domain dataset and the provided few-shot examples are from different domains (see example in Figure 10).

As also discussed in §3.4 that Codex is more familiar with SQL than Prolog, we further experiment on GeoQuery-SQL dataset (Iyer et al., 2017) which converts Prolog commands to SQL commands. We show a comparison of the two prompts in Appendix Figure 21. We found altering Prolog to SQL in prompts increases the performance dramatically, indicating aligning the expression of prompts with pre-training corpus can be another effective way of prompt engineering.

## 4 Analysis

### 4.1 How does SYMGEN compare with data augmentation methods?

For training a better task model for symbolic language tasks, data recombination (Jia and Liang, 2016) has been the common choice due to its compositional characteristics. We further compare SYMGEN with two competitive baselines for semantic parsing: Jia and Liang (2016) which uses an SCFG induced by domain-specific heuristics, and Andreas (2020) which compositionally reuses pre-

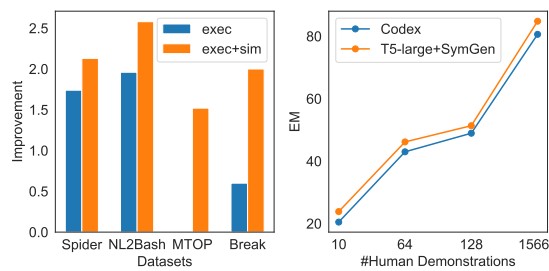

Figure 4: (a) Comparison of different verification methods. We show improvement over the baseline which directly takes the answer with maximum log-probability as output without verification; (b) Results for Codex with in-context learning and T5-large with SYMGEN using different numbers of human annotations on MTOP dataset.

viously observed sequence fragments in novel environments. We generate 1,000 instances for each method and report the results in Table 5. We can see SYMGEN provides a larger boost, especially in the few-shot setting, where Andreas (2020) failed due to the lack of initial seed data.

### 4.2 How does the verification method affect performance?

We now investigate the effectiveness of the verification method discussed in §2.2. Figure 4 (a) shows various answer verification methods, compared with picking the top-likelihood candidate without verification. We observed that verifying based on agreements of self-generated candidates ($\text{sim}(\cdot, \cdot)$) surpasses the without-verification baseline, and also improves answer quality on all the tasks more than simply checking grammar correctness ($\text{exec}(\cdot)$). Besides answer verification, we also show filtering low-confidence questions in Table 7, where the model trained on a much smaller size of data can outperform the one trained on the original data. This further indicates that low-quality data can interfere with the training process.

### 4.3 How does a different number of human annotations affect SYMGEN?

By far we have compared the few-shot results of Codex with in-context learning and T5-large with SYMGEN using 10 human annotations. In this section, we experiment with various amounts of human annotations and report the results in Figure 4 (b). We found the gap in performance between Codex and T5-Large remains virtually unchanged, which indicates the performance gain

| Prompt Types | Spider | | MTOP | | GeoQuery | | GeoQuery-SQL | |
|---|---|---|---|---|---|---|---|---|
| | EM | EX | Templete | EM | EM | EX | EM | EX |
| Full prompt | 53.97 | 67.89 | 29.31 | 20.49 | 41.79 | 72.50 | 71.43 | 85.36 |
| - w/o natural instruction | 54.64 | 67.02 | 25.19 | 15.21 | 34.29 | 71.43 | 68.57 | 83.57 |
| - w/o symbolic knowledge | 24.47 | 26.40 | 21.12 | 13.15 | 31.07 | 45.00 | 56.43 | 67.50 |
| - w/o instruction & knowledge | 23.60 | 26.31 | 16.51 | 10.38 | 25.00 | 41.79 | 55.00 | 66.43 |
| - w/o demonstrations | 45.65 | 67.21 | 0.00 | 0.00 | 0.00 | 17.86 | 39.29 | 61.79 |

Table 4: Results of few-shot answer generation with different prompts. GeoQuery-SQL refers to converting the language of few-shot examples from the original Prolog commands in GeoQuery dataset to SQL. We found symbolic knowledge and language reformulation both play key roles in generation quality, and the effect of natural instruction varies for different symbolic languages.

| Model | Few-shot | | Full-data | |
|---|---|---|---|---|
| | EM | Exec | EM | Exec |
| T5-Large | 10.00 | 12.86 | 83.93 | 91.79 |
| + Jia and Liang (2016) | 19.29 | 25.00 | 85.36 | 92.14 |
| + Andreas (2020) | - | - | 83.21 | 91.79 |
| + SYMGEN | **36.79** | **57.86** | **87.50** | **92.86** |

Table 5: Comparison of different data augmentation methods on GeoQuery dataset. SYMGEN provides a larger boost to the performance, especially in the few-shot setting.

obtained from pipeline alternation (i.e., from in-context learning to data generation and supervised tuning in SYMGEN) maintains as the size of human annotations grows. This further proves that one can always apply SYMGEN in different real scenarios from little to relatively more annotated data.

### 4.4 Data Analysis

We further conduct statistical and human evaluations on the quality of generated data from the perspective of question diversity, answer complexity, and data pair quality, based on the generated data for MBPP in the full data setting and Spider in the few-shot setting.

**Question Diversity** We measure the question diversity of the generated data for Spider and MBPP by question length and question distribution. As shown in Figure 5 (a), we find that the questions generated by SYMGEN are distributed similarly to the original dataset with more coverage. We also find the average length of the generated questions is longer than the original dataset for Spider but similar for MBPP as shown in Appendix E.1.

**Answer Complexity** We first measure the complexity of answers based on their response lengths. For Spider, as shown in Figure 5 (b), the answers generated by SYMGEN are longer on average than

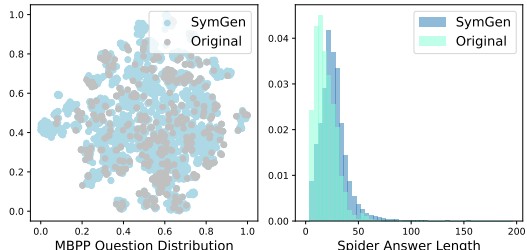

Figure 5: (a) TSNE visualization of data generated by SYMGEN (randomly sample 5000 examples) and the original data in the MBPP dataset. (b) Comparison of the length distribution of answers between the original data and SYMGEN on Spider, with the length as *x* axis and the probability density as *y* axis. More visualizations are presented in Appendix E.1.

the original dataset. Moreover, we measure the answer by their hardness, which is defined by the number of keywords following (Yu et al., 2018b). Of all the 77,828 data generated by SYMGEN, 14.15% examples are easy SQLs, 35.83% examples are of medium level, 28.13% examples are hard examples and 21.89% examples belong to extra hard SQLs. For MBPP, we found that although the generated answers have similar distribution with human-annotated data in token-level length, SYMGEN tends to generate code with more number of rows compared to human-annotated data (see Figure 8 in the Appendix). This indicates that SYMGEN can generate answers in different complexity levels, especially harder ones compared to the original human-annotated data.

**Human Evaluation of Data-pairs** In order to evaluate the quality of generated data, we also present human evaluations on the data-pair quality of generated Spider and MBPP datasets. We randomly sample 100 examples from SYMGEN for both datasets and manually review the sampled

data. We find 81 and 79 examples are correct for MBPP and Spider, respectively. Apart from that, we also find that SYMGEN generates more operators such as *julianday, union* in SQL compared to the original dataset, and the generated questions covered a wide range of data structures including *dict, list, and queue* for MBPP.

However, there are mainly three issues that exist in the data generated by SYMGEN in both MBPP and Spider. First, SYMGEN may generate ambiguous and under-specified questions (examples in Appendix E.3). Secondly, the answers sometimes can be meaninglessly complex. In Spider, SYMGEN tends to generate SQL queries with multiple JOIN clauses, therefore making the response sequences longer compared to the original dataset. Similarly. the generated Python codes tend to use *for-loop* and recursion instead of the built-in functions of Python (*e.g. max, min*). Thirdly, it can be difficult to verify the correctness of the generated answers based on either the original databases in Spider or the test cases that are generated along with Python solutions for MBPP. A quarter of the generated SQL queries have empty execution results on the original databases of Spider and more than 10% of the generated python codes have wrong test cases. We hope these could help to shed light on possible improvements for future works.

## 5 Related Work

### 5.1 Prompting LLMs

In recent years, large pre-trained language models (LLMs) have shown promising performance on zero-shot and few-shot learning tasks by prompt-based in-context learning (Radford et al., 2019; Brown et al., 2020, *inter alia*). By explicitly curating to include code (programming language) into pre-training corpora (Wang, 2021; Chen et al., 2021; Chowdhery et al., 2022, *inter alia*), LLMs exhibit surprising ability in symbolic tasks such as semantic parsing (Shin and Van Durme, 2022) and code generation (Austin et al., 2021; Poesia et al., 2021; Rajkumar et al., 2022). Nevertheless, prompt-based inference with LLMs suffers from several problems including low inference efficiency and expensive deployment cost. In this work, we employ LLMs as data generators rather than direct inferencer, which generate supervised data with minimal human effort to improve the performance of much smaller models for efficient inference on downstream tasks.

### 5.2 Data Generation

Data generation is an alternative to data augmentation by creating entirely new examples instead of combining original ones (Jia and Liang, 2016; Andreas, 2020, *inter alia*) (see Appendix F for details). Conventional approaches adopt fine-tuned generative models (Zhong et al., 2020b; Guo et al., 2021; Wang et al., 2021a, *inter alia*) as input generators, with a semantic parser (e.g., PCFG grammar) for sampling symbolic outputs. Considering the difficulty in designing grammar to sample useful symbolic forms in complex domains, Yang et al. (2022) assumes access to an unlabeled corpus of symbolic language, which is represented in canonical forms, and simulates natural language inputs via LLMs. In comparison, we explore directly generating symbolic forms as well as natural languages without the need to design task-specific grammars for symbolic forms or synchronous context-free grammars (SCFG) that map between canonical forms and symbolic forms. Data generation via LLM has also been explored under various contexts, e.g., cross-lingual semantic parsing (Rosenbaum et al., 2022), python program (Haluptzok et al., 2022), instruction generation (Wang et al., 2022), and multimodal tasks (Liu et al., 2023; Pi et al., 2023), in contrast, we aim to unify the data generation procedure for various symbolic languages tasks. Furthermore, for simple classification tasks, it has been found a smaller model trained on data generated with a few or even zero human demonstrations can achieve better performance than the LLMs (Schick and Schütze, 2021; Meng et al., 2022; Ye et al., 2022b,a; Gao et al., 2023). This work fills in the gap by exploring such an approach to complex symbolic language tasks.

## 6 Conclusion

In this work, we treat LLMs as data generators rather than task inferencer for complex symbolic language tasks, with the generated data being used to train much affordable model for depolyment and inference. We demonstrate that a 1%-sized model trained under SYMGEN can achieve superior performance to the LLM inferencers. We especially show the effectiveness in low-resource scenarios, which is a common situation for symbolic language tasks due to the annotation-expensive characteristics. Additionally, we also reveal the possibility of obtaining a well-performed task model through SYMGEN even without any human annotations.

## Limitations

This work is based on prompting and in-context learning with informative prompts for symbolic data generation. However, the information that can be packed into the prompt is hard limited by the prompt length, as language models are created and trained only to handle sequences of a certain length. The problem becomes more acute for symbolic languages with complex grammar and is rarely seen by the LLMs during the pre-training stage. Possible solutions are internalizing the grammar knowledge into the output rather than input through constrained decoding algorithms (Scholak et al., 2021; Wu et al., 2021; Shin et al., 2021; Shin and Van Durme, 2022), identifying limited relevant documentation when generating data (Agarwal et al., 2020; Zhou et al., 2022), or improving the architectures of LLMs to handle long inputs (Katharopoulos et al., 2020; Peng et al., 2020; Press et al., 2021). In addition, alternative evaluation metrics such as tree edit distances or Smatch (Cai and Knight, 2013) can be employed to reflect the similarity between two symbolic languages when execution is impractical.

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

## A Datasets

**Spider** The Spider dataset (Yu et al., 2018b) is a multi-domain and cross-database dataset for text-to-SQL parsing. There are 7,000 examples for training and 1,034 for development. We report performance on the development set as noted by Rajkumar et al. (2022) that evaluating on the held-out test set risks inadvertently leaking them for retraining of Codex.[1] We determine model performance based on surface form Exact Set Match (EM; Yu et al. (2018b)) and test-suite Execution accuracy (EX; Zhong et al. (2020a)) which extends execution to multiple database instances per SQL schema to provide the best approximation of semantic accuracy.

**NL2Bash** The NL2Bash dataset (Lin et al., 2018) aims to translate natural language to bash commands. There are 8,090 examples for training and 609 for development. Because it is difficult to execute bash commands in a sandbox, we evaluate the a bash command by parsing and tokenizing with bashlex[2], and calculating token-level BLEU-4 score between commands as the estimation of execution result similarity. Following Lin et al. (2018), commands are evaluated with character-level BLEU-4 score.

**MBPP** The MBPP dataset (Austin et al., 2021) is a python programming task, where text description is mapped to python program containing multiple lines. MBPP consists of 974 examples, with 500 of them used for testing and the rest for training or few-shot prompting. We evaluate with execution accuracy (EX), where a program is considered as passing if all three associated test cases

are correct. We don't include surface-level metrics (e.g., BLEU) as semantically identical programs can potentially have very low n-gram overlap (e.g., identifier renaming) (Austin et al., 2021).

**GeoQuery** The GeoQuery dataset (Zelle and Mooney, 1996) contains human-authored questions paired with prolog logic programming language about U.S. geography, with 600 examples for training and 280 for testing. We report Exact Match (EM) and Execution (EX) accuracy by running with SWI-Prolog[3] and pyswip[4].

**MTOP** MTOP (Li et al., 2021) is a semantic parsing dataset, focused on multilingual task-oriented dialogues, where commands are mapped to complex nested queries across 11 domains. Similar to previous work (Pasupat et al., 2021), we use the English subset, which contains 15,667 training examples and 2,235 development set examples. We evaluate with Exact Match (EM), i.e., whether the prediction string is identical to the reference string, and Template Accuracy where the query tokens are discarded (e.g., the template of [IN:A [SL:B text]] is [IN:A [SL:B]]).

**Break** The Break dataset Wolfson et al. (2020) contains complex natural language questions sampled from 10 QA datasets, and they are decomposed into an ordered list of atomic steps. We use the low-level subset, which contains 44,321 training examples and 8,000 development set examples. We randomly sample 1,000 examples to construct a new development set for evaluation. We evaluate model performance with LF-EM (Hasson and Berant, 2021), which is proposed as an improvement to Exact Match (EM) to measure whether two meaning representations are semantically equivalent.

## B Implementation Details

For prompting or in-context learning with Codex, we use code-davinci-002 and a maximum context size of 7000. For all the tasks, we set the temperature to 0.8 and the number of samplings to 30 for answer generation. When generating questions, we construct initial 200 prompts by randomly selecting in-context examples[5] and use the mixture of temperature (i.e., 0.6, 0.8, and 1) with a number of

---

[1] Unless otherwise mentioned, we also report the results on the development set for the other datasets.

[2] https://pypi.org/project/bashlex/

[3] https://www.swi-prolog.org/

[4] https://github.com/yuce/pyswip

[5] In few-shot settings, we random sample permutation of all the examples to infuse diversity.

samplings of 100 to generate at most 60k questions. For Spider, we generate 200 questions for each of the 140 databases in the training set, which results in at most 84k data pairs using three temperatures. We set the number of shots to 10 in the few-shot setting. In the full-data setting, as found by prior work that including similar exemplars helps in answer prediction (Liu et al., 2022; Wu et al., 2022; Ye et al., 2023), we use all-mpnet-base-v2 (Song et al., 2020)[6] to encode questions and Faiss[7] to search similar examples. We truncate the number of in-context examples based on the maximum context size and order the examples from least to most based on similarity score.

We mainly use T5-large (770M) and T5-3B (Raffel et al., 2020) as task models for all the datasets. For MBPP (python) dataset, we find the original tokenizers of T5 is based on SentencePiece and would remove the indentations and blankspaces in the codes when doing tokenization, and therefore would influence the execution of Python program when generating the code string. Based on this reason, we use CodeT5-large (770M; Wang et al. 2021b) on MBPP dataset.

For training T5, we adopt the setting from Xie et al. (2022), where we use a batch size of 32, an Adafactor (Duchi et al., 2011) optimizer for T5-large, an AdamW (Loshchilov and Hutter, 2018) optimizer for T5-3B, a learning rate of 5e-5, a linear learning rate decay and a maximum number of training epochs of 50 with early-stopping patience of 5. In the full-data setting, we use the strategy of first tuning on the mixture of synthesized and human-annotated data, then continue tuning it with only the human annotation data. We find this two-stage training performs better than the importance-weighted loss (see Appendix C for details).

## C Training Strategy

We compare the training strategies when we have both full human annotated data and generated data in Table 6. We can see the two-stage training procedure that first trains on the mixture on both datasets and then solely on human annotated data outperforms the weighted training baselines.

---

[6]https://huggingface.co/sentence-transformers/all-mpnet-base-v2

[7]https://github.com/facebookresearch/faiss

|  | MTOP | NL2bash | Break | Spider |
|---|---|---|---|---|
| #Data (Gold+Syn.) | 15k+36k | 8k+47k | 45k+41k | 7k+82k |
| Gold | 83.04 | 65.95 | 52.70 | 64.12 |
| Mix 1:1 | 81.88 | 62.88 | 51.60 | 68.38 |
| Mix 1:3 | 83.27 | 66.50 | 53.10 | 68.57 |
| Mix 1:1 → Gold | **84.83** | **67.17** | **54.50** | **69.63** |

Table 6: Training strategies to use full human-annotated data and synthetic data using T5-Large. The two-stage training strategy (last row) performs better than the importance-weighted loss.

## D Question Verification Results

We measure the quality of a question through answer consistency, where more generated answers are semantically equivalent means the question is less ambiguous and considered as high quality. We show the effect of the threshold used to filter ambiguous question in Table 7. We can see the model trained on a much smaller size of data can outperform the one trained on original data, indicating low quality data can interfere with the training process.

| Thre. | NL2Bash | | MTOP | | Break | |
|---|---|---|---|---|---|---|
|  | #data | CBLEU | #data | EM | #data | LF-EM |
| $T=0$ | 58k | 56.83 | 40k | 23.13 | 56k | 29.90 |
| $T=3$ | 46k | 56.49 | 34k | **23.85** | 41k | **30.30** |
| $T=5$ | 39k | **57.83** | 27k | 22.10 | 29k | 28.30 |

Table 7: Results on filtering generated questions with varying threshold $T$ on few-shot setting and training T5-large. We found filtering questions that have low-confidence answers results in a smaller dataset but improves model performance.

## E Data Analysis

### E.1 Question Diversity

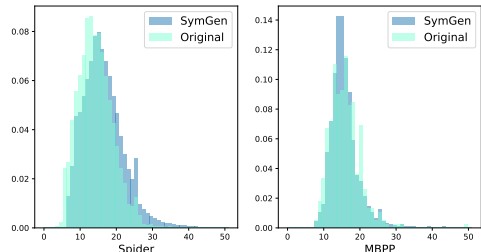

Figure 6: Comparison on token-level length distribution of the questions on Spider and MBPP.

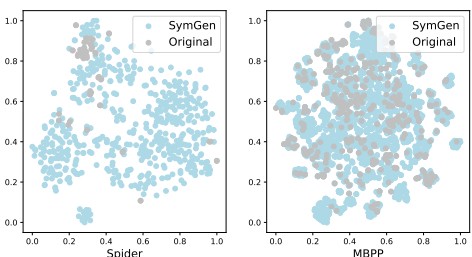

Figure 7: Comparison on the distribution of the questions' embedding (obtained by SBERT) in Spider (randomly sample one database) and MBPP from SYMGEN (randomly sample 5000) and the original datasets.

### E.2 Response Complexity

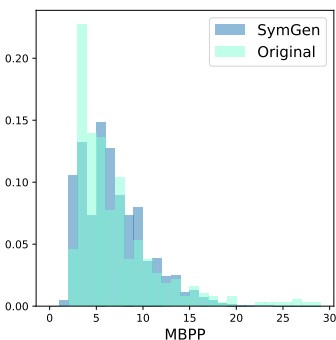

Figure 8: Comparison on row-level lengths of the answers between SYMGEN generated data and the original MBPP dataset.

### E.3 Human Evaluation on Data Pair Quality

In this section we present some typical examples of the question ambiguity and underspecification problems in data generation by SYMGEN. The generated questions may be underspecified and ambiguous, even sometimes unreasonable, therefore influencing the generation of corresponding answers. Some examples are presented as follows.

```
# Spider
(1) How to modify table Customer_Orders
such that for every row, system will
automatically insert a row to table
Customer_ord
(2) List the name and the year in which
the party was first elected for the
parties that have been elected in at
least 2 counties.

# MBPP
(1) Write a function to convert a string
```

to a list. (Didn't mention in character-
-level or word-level)
(2) Write a function to split the given
comma-separated values of list into two
lists. (Didn't tell the rule to split
the list)

## F    Related Work on Data Augmentation

There is a large body of research on general data augmentation (Feng et al., 2021), which assume the outputs remain unchanged. For symbolic-language prediction tasks, instead of holding outputs fixed, we would like to apply simultaneous transformations to inputs and outputs to increase the coverage of output structures. Data recombination method (Jia and Liang, 2016; Andreas, 2020; Akyürek et al., 2020; Guo et al., 2020; Qiu et al., 2022a) along both dimensions of inputs and outputs are proposed, where different fragments of input and output from different examples are re-combined to create hard (Jia and Liang, 2016; Andreas, 2020; Akyürek et al., 2020; Qiu et al., 2022a) or soft (Guo et al., 2020) augmented examples. Yu et al. (2018a, 2020) follow the same spirit and use a hand-crafted SCFG grammar to generate new parallel data. However, rule-based heuristics or a large pool of seed examples are needed to induce the grammar.

## G    Prompt Examples

```
CREATE TABLE "Rooms" ("RoomId" TEXT PRIMARY KEY, "roomName" TEXT, "beds" INTEGER, "bedType" TEXT, "maxOccupancy" INTEGER, "basePrice" INTEGER, "decor"
        TEXT)
/*
3 example rows from table Rooms:
SELECT * FROM Rooms LIMIT 3;
RoomId            roomName  beds bedType  maxOccupancy  basePrice  decor
  HBB Harbinger but bequest    1   Queen             2        100 modern
  TAA    Thrift and accolade   1  Double             2         75 modern
  RTE    Riddle to exculpate   2   Queen             4        175 rustic
*/

CREATE TABLE "Reservations" ("Code" INTEGER PRIMARY KEY, "Room" TEXT, "CheckIn" TEXT, "CheckOut" TEXT, "Rate" REAL, "LastName" TEXT, "FirstName" TEXT,
        "Adults" INTEGER, "Kids" INTEGER, FOREIGN KEY (Room) REFERENCES Rooms(RoomId))
/*
3 example rows from table Reservations:
SELECT * FROM Reservations LIMIT 3;
 Code Room   CheckIn  CheckOut    Rate LastName FirstName Adults Kids
60313 CAS 28-OCT-10 30-OCT-10 218.75    SLONE    LARITA      1    1
81473 RND 01-FEB-10 02-FEB-10 127.50  EVERITT       YUK      1    1
35546 TAA 19-SEP-10 24-SEP-10  67.50      YUK       TIM      1    0
*/

-- Write a question that can be answered based on the above tables.
-- Question: List the type of bed and name of all traditional rooms.

 ** EXAMPLE SEPARATOR **

CREATE TABLE "department" ("Department_ID" int, "Name" text, "Creation" text, "Ranking" int, "Budget_in_Billions" real, "Num_Employees" real, PRIMARY
        KEY ("Department_ID"))
/*
3 example rows from table department:
SELECT * FROM department LIMIT 3;
 Department_ID            Name Creation  Ranking  Budget_in_Billions  Num_Employees
            7         Commerce     1903        7                 6.2        36000.0
            3          Defense     1947        3               439.3      3000000.0
           15 Homeland Security    2002       15                44.6       208000.0
*/

CREATE TABLE "head" ("head_ID" int, "name" text, "born_state" text, "age" real, PRIMARY KEY ("head_ID"))
/*
3 example rows from table head:
SELECT * FROM head LIMIT 3;
 head_ID          name born_state  age
       8   Nick Faldo California 56.0
       7 Stewart Cink    Florida 50.0
       5 Jeff Maggert   Delaware 53.0
*/

CREATE TABLE "management" ("department_ID" int, "head_ID" int, "temporary_acting" text, PRIMARY KEY ("Department_ID","head_ID"), FOREIGN KEY ("
        Department_ID") REFERENCES `department`("Department_ID"), FOREIGN KEY ("head_ID") REFERENCES `head`("head_ID"))
/*
3 example rows from table management:
SELECT * FROM management LIMIT 3;
 department_ID  head_ID temporary_acting
            7        3               No
           15        4              Yes
           11       10               No
*/

-- Write a question that can be answered based on the above tables.
-- Question:
```

Figure 9: Example prompt for generating questions for Spider, only single in-context example is shown for illustration.

```
CREATE TABLE "Rooms" ("RoomId" TEXT PRIMARY KEY, "roomName" TEXT, "beds" INTEGER, "bedType" TEXT, "maxOccupancy" INTEGER, "basePrice" INTEGER, "decor"
        TEXT)
/*
3 example rows from table Rooms:
SELECT * FROM Rooms LIMIT 3;
RoomId          roomName  beds bedType  maxOccupancy  basePrice  decor
   HBB Harbinger but bequest    1   Queen            2        100 modern
   TAA   Thrift and accolade    1  Double            2         75 modern
   RTE   Riddle to exculpate    2   Queen            4        175 rustic
*/

CREATE TABLE "Reservations" ("Code" INTEGER PRIMARY KEY, "Room" TEXT, "CheckIn" TEXT, "CheckOut" TEXT, "Rate" REAL, "LastName" TEXT, "FirstName" TEXT,
        "Adults" INTEGER, "Kids" INTEGER, FOREIGN KEY (Room) REFERENCES Rooms(RoomId))
/*
3 example rows from table Reservations:
SELECT * FROM Reservations LIMIT 3;
 Code Room   CheckIn  CheckOut   Rate LastName FirstName  Adults  Kids
60313  CAS 28-OCT-10 30-OCT-10 218.75    SLONE    LARITA       1     1
81473  RND 01-FEB-10 02-FEB-10 127.50  EVERITT       YUK       1     1
35546  TAA 19-SEP-10 24-SEP-10  67.50      YUK       TIM       1     0
*/

-- Using valid SQLite, answer the following questions for the tables provided above.
-- Question: List the type of bed and name of all traditional rooms.
SELECT roomName ,  bedType FROM Rooms WHERE decor = "traditional";

 ** EXAMPLE SEPARATOR **

CREATE TABLE "department" ("Department_ID" int, "Name" text, "Creation" text, "Ranking" int, "Budget_in_Billions" real, "Num_Employees" real, PRIMARY
        KEY ("Department_ID"))
/*
3 example rows from table department:
SELECT * FROM department LIMIT 3;
 Department_ID       Name Creation  Ranking  Budget_in_Billions  Num_Employees
             1      State     1789        1                9.96        30266.0
             2   Treasury     1789        2               11.10       115897.0
             3    Defense     1947        3              439.30      3000000.0
*/

CREATE TABLE "head" ("head_ID" int, "name" text, "born_state" text, "age" real, PRIMARY KEY ("head_ID"))
/*
3 example rows from table head:
SELECT * FROM head LIMIT 3;
 head_ID         name born_state  age
       1  Tiger Woods    Alabama 67.0
       2 Sergio Garcia California 68.0
       3    K. J. Choi    Alabama 69.0
*/

CREATE TABLE "management" ("department_ID" int, "head_ID" int, "temporary_acting" text, PRIMARY KEY ("Department_ID","head_ID"), FOREIGN KEY ("
        Department_ID") REFERENCES `department`("Department_ID"), FOREIGN KEY ("head_ID") REFERENCES `head`("head_ID"))
/*
3 example rows from table management:
SELECT * FROM management LIMIT 3;
 department_ID  head_ID temporary_acting
            2        5              Yes
           15        4              Yes
            2        6              Yes
*/

-- Using valid SQLite, answer the following questions for the tables provided above.
-- Question: What are the names of the heads who manage the department with ID 15?
SELECT
```

Figure 10: Example prompt for generating SQL queries for Spider, only single in-context example is shown for illustration.

```
Translate the natural language description to bash commands.

Natural Language: Recursively removes all files and folders named '.svn' in a current folder, handling content of removed folder before folder inself.

Natural Language: find all executable files in /home directory.

Natural Language: Locate files that reside in the /u/bill directory tree and were last accessed between 2 and 6 minutes ago

Natural Language: Search the current directory tree for files whose names match regular expression '.*packet.*', ignoring the case

Natural Language: List all the empty files in thecurrent directory only.

Natural Language: Find all files under current directory whose status was changed less than 3 days ago and show last 5 lines of output

Natural Language: Find files that were modified more than 7 days ago and archive them

Natural Language: Set variable 'file' to the base name of first argument to script or function, that is the part following the last slash.

Natural Language: Connect to host "$USER_AT_HOST" in master mode in the background without executing any commands and set the ControlPath to "
        $SSHSOCKET"

Natural Language: Print input "your, text, here" formatted to fit 70 characters per line breaking at spaces

Natural Language:
```

Figure 11: Example prompt for generating questions for NL2Bash.

```
Translate the natural language description to bash commands.

Natural Language: Recursively removes all files and folders named '.svn' in a current folder, handling content of removed folder before folder inself.
Bash commands: find . -depth -name .svn -exec rm -fr {} \;

Natural Language: find all executable files in /home directory.
Bash commands: find  /home -type f -perm /a=x

Natural Language: Locate files that reside in the /u/bill directory tree and were last accessed between 2 and 6 minutes ago
Bash commands: find /u/bill -amin +2 -amin -6

Natural Language: Search the current directory tree for files whose names match regular expression '.*packet.*', ignoring the case
Bash commands: find . -iregex ".*packet.*"

Natural Language: List all the emptry files in thecurrent directory only.
Bash commands: find . -maxdepth 1 -empty

Natural Language: Find all files under current directory whose status was changed less than 3 days ago and show last 5 lines of output
Bash commands: find . -type f -ctime -3 | tail -n 5

Natural Language: Find files that were modified more than 7 days ago and archive them
Bash commands: find . -type f -mtime +7 | xargs tar -cvf `date '+%d%m%Y'_archive.tar`

Natural Language: Set variable 'file' to the base name of first argument to script or function, that is the part following the last slash.
Bash commands: file=`basename "$1"`

Natural Language: Connect to host "$USER_AT_HOST" in master mode in the background without executing any commands and set the ControlPath to "
        $SSHSOCKET"
Bash commands: ssh -M -f -N -o ControlPath="$SSHSOCKET" "$USER_AT_HOST"

Natural Language: Print input "your, text, here" formatted to fit 70 characters per line breaking at spaces
Bash commands: echo 'your, text, here' | fold -sw 70

Natural Language: Find files with names that start with 'input' and end with a single character 'a' or 'b' in the current directory and all its
        subdirectories
Bash commands:
```

Figure 12: Example prompt for generating bash commands for NL2Bash.

```
Translate the natural language instructions to Python codes.

Natural Language Instruction for Python Code: Write a function to find squares of individual elements in a list using lambda function.

Natural Language Instruction for Python Code: Write a function to find all words which are at least 4 characters long in a string by using regex.

Natural Language Instruction for Python Code: Write a python function to find the minimum number of rotations required to get the same string.

Natural Language Instruction for Python Code: Write a python function to check whether the two numbers differ at one bit position only or not.

Natural Language Instruction for Python Code: Write a python function to identify non-prime numbers.

Natural Language Instruction for Python Code: Write a function to find the largest integers from a given list of numbers using heap queue algorithm.

Natural Language Instruction for Python Code: Write a function to get the n smallest items from a dataset.

Natural Language Instruction for Python Code: Write a function to find the number of ways to fill it with 2 x 1 dominoes for the given 3 x n board.

Natural Language Instruction for Python Code: Write a function to find the minimum cost path to reach (m, n) from (0, 0) for the given cost matrix cost
        [][] and a position (m, n) in cost[][].

Natural Language Instruction for Python Code: Write a function to find the similar elements from the given two tuple lists.

Natural Language Instruction for Python Code:
```

Figure 13: Example prompt for generating question descriptions for MBPP.

```python
"""
Write a python function to check whether the word is present in a given sentence or not.
"""
def is_Word_Present(sentence,word):
    s = sentence.split(" ")
    for i in s:
        if (i == word):
            return True
    return False
# 3 test cases
assert is_Word_Present("machine learning","machine") == True
assert is_Word_Present("easy","fun") == False
assert is_Word_Present("python language","code") == False

"""
Write a function to find the cumulative sum of all the values that are present in the given tuple list.
"""
def cummulative_sum(test_list):
  res = sum(map(sum, test_list))
  return (res)
# 3 test cases
assert cummulative_sum([(1, 3), (5, 6, 7), (2, 6)]) == 30
assert cummulative_sum([(2, 4), (6, 7, 8), (3, 7)]) == 37
assert cummulative_sum([(3, 5), (7, 8, 9), (4, 8)]) == 44

"""
Write a python function to find the average of a list.
"""
def Average(lst):
    return sum(lst) / len(lst)
# 3 test cases
assert Average([15, 9, 55, 41, 35, 20, 62, 49]) == 35.75
assert Average([4, 5, 1, 2, 9, 7, 10, 8]) == 5.75
assert Average([1,2,3]) == 2

...... (Some in-context examples omitted here for simplicity)

"""
Write a function to create a list taking alternate elements from another given list.
"""
def alternate_elements(list1):
    result=[]
    for item in list1[::2]:
        result.append(item)
    return result
# 3 test cases
assert alternate_elements(["red", "black", "white", "green", "orange"])==['red', 'white', 'orange']
assert alternate_elements([2, 0, 3, 4, 0, 2, 8, 3, 4, 2])==[2, 3, 0, 8, 4]
assert alternate_elements([1, 2, 3, 4, 5, 6, 7, 8, 9, 10])==[1,3,5,7,9]

"""
Write a function to find the minimum total path sum in the given triangle.
"""
def min_sum_path(A):
        memo = [None] * len(A)
        n = len(A) - 1
        for i in range(len(A[n])):
                memo[i] = A[n][i]
        for i in range(len(A) - 2, -1,-1):
                for j in range( len(A[i])):
                        memo[j] = A[i][j] + min(memo[j],
                                                    memo[j + 1])
        return memo[0]
# 3 test cases
assert min_sum_path([[ 2 ], [3, 9 ], [1, 6, 7 ]]) == 6
assert min_sum_path([[ 2 ], [3, 7 ], [8, 5, 6 ]]) == 10
assert min_sum_path([[ 3 ], [6, 4 ], [5, 2, 7 ]]) == 9

"""
Write a python function to count occurences of a character in a repeated string.
"""
def count_Char(str,x):
    count = 0
    for i in range(len(str)):
        if (str[i] == x) :
            count += 1
    n = 10
    repititions = n // len(str)
    count = count * repititions
    l = n % len(str)
    for i in range(l):
        if (str[i] == x):
            count += 1
    return count
# 3 test cases
assert count_Char("abcac",'a') == 4
assert count_Char("abca",'c') == 2
assert count_Char("aba",'a') == 7

"""
Write a function to find the difference between two lists.
"""
```

Figure 14: Example prompt for generating python program queries for MBPP.

IN:GET: MESSAGE, WEATHER, ALARM, INFO_RECIPES, STORIES_NEWS, REMINDER, RECIPES, EVENT, CALL_TIME, LIFE_EVENT, INFO_CONTACT, CONTACT, TIMER,
        REMINDER_DATE_TIME, AGE, SUNRISE, EMPLOYER, EDUCATION_TIME, JOB, AVAILABILITY, CATEGORY_EVENT, CALL, EMPLOYMENT_TIME, CALL_CONTACT, LOCATION,
        TRACK_INFO_MUSIC, SUNSET, MUTUAL_FRIENDS, UNDERGRAD, REMINDER_LOCATION, ATTENDEE_EVENT, MESSAGE_CONTACT, REMINDER_AMOUNT, DATE_TIME_EVENT,
        DETAILS_NEWS, EDUCATION_DEGREE, MAJOR, CONTACT_METHOD, LIFE_EVENT_TIME, LYRICS_MUSIC, AIRQUALITY, LANGUAGE, GENDER, GROUP | IN:SEND: MESSAGE | IN
        :SET: UNAVAILABLE, RSVP_YES, AVAILABLE, DEFAULT_PROVIDER_MUSIC, RSVP_INTERESTED, DEFAULT_PROVIDER_CALLING, RSVP_NO | IN:DELETE: REMINDER, ALARM,
        TIMER, PLAYLIST_MUSIC | IN:CREATE: ALARM, REMINDER, CALL, PLAYLIST_MUSIC, TIMER | IN:QUESTION: NEWS, MUSIC | IN:PLAY: MUSIC, MEDIA | IN:END: CALL
        | IN:IGNORE: CALL | IN:UPDATE: CALL, REMINDER_DATE_TIME, REMINDER_TODO, TIMER, METHOD_CALL, ALARM, REMINDER_LOCATION, REMINDER | IN:PAUSE: MUSIC
        , TIMER | IN:ANSWER: CALL | IN:SNOOZE: ALARM | IN:IS: TRUE_RECIPES | IN:REMOVE: FROM_PLAYLIST_MUSIC | IN:ADD: TIME_TIMER, TO_PLAYLIST_MUSIC | IN:
        SHARE: EVENT | IN:PREFER:  | IN:START: SHUFFLE_MUSIC | IN:SILENCE: ALARM | IN:SWITCH: CALL | IN:SUBTRACT: TIME_TIMER | IN:PREVIOUS: TRACK_MUSIC |
        IN:HOLD: CALL | IN:SKIP: TRACK_MUSIC | IN:LIKE: MUSIC | IN:RESTART: TIMER | IN:RESUME: TIMER, CALL, MUSIC | IN:MERGE: CALL | IN:REPLAY: MUSIC |
        IN:LOOP: MUSIC | IN:STOP: MUSIC, SHUFFLE_MUSIC | IN:UNLOOP: MUSIC | IN:CANCEL: MESSAGE, CALL | IN:REWIND: MUSIC | IN:REPEAT: ALL_MUSIC,
        ALL_OFF_MUSIC | IN:FAST: FORWARD_MUSIC | IN:DISLIKE: MUSIC | IN:DISPREFER:  | IN:HELP: REMINDER | IN:FOLLOW: MUSIC
SL:CONTACT: , ADDED, RELATED, REMOVED, METHOD | SL:TYPE: CONTENT, RELATION, CONTACT | SL:RECIPIENT:  | SL:LOCATION:  | SL:DATE: TIME | SL:ORDINAL:  |
        SL:CONTENT: EXACT | SL:RECIPES: ATTRIBUTE, DISH, COOKING_METHOD, INCLUDED_INGREDIENT, TYPE, UNIT_NUTRITION, EXCLUDED_INGREDIENT, DIET,
        UNIT_MEASUREMENT, TYPE_NUTRITION, MEAL, RATING, QUALIFIER_NUTRITION, SOURCE, CUISINE | SL:PERSON: REMINDED | SL:TODO:  | SL:NEWS: TYPE, CATEGORY,
        TOPIC, REFERENCE, SOURCE | SL:SENDER:  | SL:MUSIC: TYPE, ARTIST_NAME, PLAYLIST_TITLE, TRACK_TITLE, PROVIDER_NAME, GENRE, ALBUM_TITLE, RADIO_ID,
        ALBUM_MODIFIER, REWIND_TIME, PLAYLIST_MODIFIER | SL:NAME: APP | SL:WEATHER: ATTRIBUTE, TEMPERATURE_UNIT | SL:CATEGORY: EVENT | SL:METHOD: TIMER,
        RETRIEVAL_REMINDER, RECIPES | SL:LIFE: EVENT | SL:AMOUNT:  | SL:EMPLOYER:  | SL:PERIOD:  | SL:EDUCATION: DEGREE | SL:TITLE: EVENT | SL:TIMER:
        NAME | SL:JOB:  | SL:PHONE: NUMBER | SL:ATTRIBUTE: EVENT | SL:ALARM: NAME | SL:SCHOOL:  | SL:SIMILARITY:  | SL:GROUP:  | SL:AGE:  | SL:ATTENDEE:
        EVENT,  | SL:USER: ATTENDEE_EVENT | SL:MAJOR:  | SL:GENDER:

Translate the natural language description to logical form with the above arguments.

Natural Language: Every day my alarm is set for what time?

Natural Language: top news stories

Natural Language: should I bring the plants in tonight

Natural Language: Resume the timer in 10 seconds

Natural Language: Message Ben to see if he can come to my birthday party

Natural Language: Do I have a friend that works in Los Angeles?

Natural Language: give me the news update for around the world.

Natural Language: Set the sleep timer for 10 minutes

Natural Language: i need to change my podiatrist appointment reminder to 5pm instead of 5:30

Natural Language: take 11 minutes off the timer

Natural Language:

Figure 15: Example prompt for generating questions for MTOP.

```
IN:GET: MESSAGE, WEATHER, ALARM, INFO_RECIPES, STORIES_NEWS, REMINDER, RECIPES, EVENT, CALL_TIME, LIFE_EVENT, INFO_CONTACT, CONTACT, TIMER,
    REMINDER_DATE_TIME, AGE, SUNRISE, EMPLOYER, EDUCATION_TIME, JOB, AVAILABILITY, CATEGORY_EVENT, CALL, EMPLOYMENT_TIME, CALL_CONTACT, LOCATION,
    TRACK_INFO_MUSIC, SUNSET, MUTUAL_FRIENDS, UNDERGRAD, REMINDER_LOCATION, ATTENDEE_EVENT, MESSAGE_CONTACT, REMINDER_AMOUNT, DATE_TIME_EVENT,
    DETAILS_NEWS, EDUCATION_DEGREE, MAJOR, CONTACT_METHOD, LIFE_EVENT_TIME, LYRICS_MUSIC, AIRQUALITY, LANGUAGE, GENDER, GROUP | IN:SEND: MESSAGE | IN
    :SET: UNAVAILABLE, RSVP_YES, AVAILABLE, DEFAULT_PROVIDER_MUSIC, RSVP_INTERESTED, DEFAULT_PROVIDER_CALLING, RSVP_NO | IN:DELETE: REMINDER, ALARM,
    TIMER, PLAYLIST_MUSIC | IN:CREATE: ALARM, REMINDER, CALL, PLAYLIST_MUSIC, TIMER | IN:QUESTION: NEWS, MUSIC | IN:PLAY: MUSIC, MEDIA | IN:END: CALL
    | IN:IGNORE: CALL | IN:UPDATE: CALL, REMINDER_DATE_TIME, REMINDER_TODO, TIMER, METHOD_CALL, ALARM, REMINDER_LOCATION, REMINDER | IN:PAUSE: MUSIC
    , TIMER | IN:ANSWER: CALL | IN:SNOOZE: ALARM | IN:IS: TRUE_RECIPES | IN:REMOVE: FROM_PLAYLIST_MUSIC | IN:ADD: TIME_TIMER, TO_PLAYLIST_MUSIC | IN:
    SHARE: EVENT | IN:PREFER:  | IN:START: SHUFFLE_MUSIC | IN:SILENCE: ALARM | IN:SWITCH: CALL | IN:SUBTRACT: TIME_TIMER | IN:PREVIOUS: TRACK_MUSIC
    | IN:HOLD: CALL | IN:SKIP: TRACK_MUSIC | IN:LIKE: MUSIC | IN:RESTART: TIMER | IN:RESUME: TIMER, CALL, MUSIC | IN:MERGE: CALL | IN:REPLAY: MUSIC |
    IN:LOOP: MUSIC | IN:STOP: MUSIC, SHUFFLE_MUSIC | IN:UNLOOP: MUSIC | IN:CANCEL: MESSAGE, CALL | IN:REWIND: MUSIC | IN:REPEAT: ALL_MUSIC,
    ALL_OFF_MUSIC | IN:FAST: FORWARD_MUSIC | IN:DISLIKE: MUSIC | IN:DISPREFER:  | IN:HELP: REMINDER | IN:FOLLOW: MUSIC
SL:CONTACT: , ADDED, RELATED, REMOVED, METHOD | SL:TYPE: CONTENT, RELATION, CONTACT | SL:RECIPIENT:  | SL:LOCATION:  | SL:DATE: TIME | SL:ORDINAL:  |
    SL:CONTENT: EXACT | SL:RECIPES: ATTRIBUTE, DISH, COOKING_METHOD, INCLUDED_INGREDIENT, TYPE, UNIT_NUTRITION, EXCLUDED_INGREDIENT, DIET,
    UNIT_MEASUREMENT, TYPE_NUTRITION, MEAL, RATING, QUALIFIER_NUTRITION, SOURCE, CUISINE | SL:PERSON: REMINDED | SL:TODO:  | SL:NEWS: TYPE, CATEGORY,
    TOPIC, REFERENCE, SOURCE | SL:SENDER:  | SL:MUSIC: TYPE, ARTIST_NAME, PLAYLIST_TITLE, TRACK_TITLE, PROVIDER_NAME, GENRE, ALBUM_TITLE, RADIO_ID,
    ALBUM_MODIFIER, REWIND_TIME, PLAYLIST_MODIFIER | SL:NAME: APP | SL:WEATHER: ATTRIBUTE, TEMPERATURE_UNIT | SL:CATEGORY: EVENT | SL:METHOD: TIMER,
    RETRIEVAL_REMINDER, RECIPES | SL:LIFE: EVENT | SL:AMOUNT:  | SL:EMPLOYER:  | SL:PERIOD:  | SL:EDUCATION: DEGREE | SL:TITLE: EVENT | SL:TIMER:
    NAME | SL:JOB:  | SL:PHONE: NUMBER | SL:ATTRIBUTE: EVENT | SL:ALARM: NAME | SL:SCHOOL:  | SL:SIMILARITY:  | SL:GROUP:  | SL:AGE:  | SL:ATTENDEE:
    EVENT,  | SL:USER: ATTENDEE_EVENT | SL:MAJOR:  | SL:GENDER:

Translate the natural language description to logical form with the above arguments.

Natural Language: Every day my alarm is set for what time?
Logical Form: [IN:GET_ALARM [SL:PERIOD Every day ] ]

Natural Language: top news stories
Logical Form: [IN:GET_STORIES_NEWS [SL:NEWS_REFERENCE top ] [SL:NEWS_TYPE news stories ] ]

Natural Language: should I bring the plants in tonight
Logical Form: [IN:GET_WEATHER [SL:WEATHER_ATTRIBUTE plants ] [SL:DATE_TIME in tonight ] ]

Natural Language: Resume the timer in 10 seconds
Logical Form: [IN:RESUME_TIMER [SL:METHOD_TIMER timer ] [SL:DATE_TIME in 10 seconds ] ]

Natural Language: Message Ben to see if he can come to my birthday party
Logical Form: [IN:SEND_MESSAGE [SL:RECIPIENT Ben ] [SL:CONTENT_EXACT he can come to my birthday party ] ]

Natural Language: Do I have a friend that works in Los Angeles?
Logical Form: [IN:GET_CONTACT [SL:CONTACT_RELATED I ] [SL:TYPE_RELATION friend ] [SL:LOCATION Los Angeles ] ]

Natural Language: give me the news update for around the world.
Logical Form: [IN:GET_STORIES_NEWS [SL:NEWS_TYPE news ] ]

Natural Language: Set the sleep timer for 10 minutes
Logical Form: [IN:CREATE_TIMER [SL:TIMER_NAME sleep ] [SL:METHOD_TIMER timer ] [SL:DATE_TIME for 10 minutes ] ]

Natural Language: i need to change my podiatrist appointment reminder to 5pm instead of 5:30
Logical Form: [IN:UPDATE_REMINDER_DATE_TIME [SL:PERSON_REMINDED my ] [SL:TODO podiatrist appointment ] [SL:DATE_TIME to 5 pm ] [SL:DATE_TIME of 5 : 30
    ] ]

Natural Language: take 11 minutes off the timer
Logical Form: [IN:SUBTRACT_TIME_TIMER [SL:DATE_TIME 11 minutes ] [SL:METHOD_TIMER timer ] ]

Natural Language: What is my alarm set to every day.
Logical Form:
```

Figure 16: Example prompt for generating TOP-representations for MTOP.

```
Break down a question into the requisite steps for computing its answer.

Question: If both images show mainly similar-shaped orange-and-white striped fish swimming among anemone tendrils.

Question: If two seals are lying in the sand in the image on the right.

Question: If the right image shows a single dog sitting.

Question: How many large metallic items are there?

Question: who was the leader of the north during the vietnam war?

Question: What actor played in both the Trial of Michael Jackson and The Wiz?

Question: If the bed set in the left image has a pink canopy above it.

Question: Give the name of the student in the History department with the most credits.

Question: when did lil wayne first start singing?

Question: If there are no less than five dogs

Question:
```

Figure 17: Example prompt for generating questions for Break.

```
Break down a question into the requisite steps for computing its answer.

Question: If both images show mainly similar-shaped orange-and-white striped fish swimming among anemone tendrils.
Answer Steps: 1#) return fish 2#) return #1 that are similar-shaped 3#) return #2 that are orange-and-white striped 4#) return anemone tendrils 5#)
        return #3 swimming among #4 6#) return if #5 are mainly in both images

Question: If two seals are lying in the sand in the image on the right.
Answer Steps: 1#) return right image 2#) return seals in #1 3#) return sand in #1 4#) return #2 that are lying in #3 5#) return number of #4 6#) return
        if #5 is equal to two

Question: If the right image shows a single dog sitting.
Answer Steps: 1#) return the right image 2#) return dogs in #1 3#) return #2 that are sitting 4#) return number of #3 5#) return if #4 is equal to one

Question: How many large metallic items are there?
Answer Steps: 1#) return items 2#) return #1 that are large 3#) return #2 that are metallic 4#) return number of #3

Question: who was the leader of the north during the vietnam war?
Answer Steps: 1#) return north vietnam 2#) return leader of #1 3#) return the vietnam war 4#) return #2 during #3

Question: What actor played in both the Trial of Michael Jackson and The Wiz?
Answer Steps: 1#) return Trial of Michael Jackson 2#) return The Wiz 3#) return actor of both #1 and #2

Question: If the bed set in the left image has a pink canopy above it.
Answer Steps: 1#) return left image 2#) return bed set in #1 3#) return canopy in #1 4#) return #3 that is pink 5#) return #4 that is above #2 6#)
        return number of #5 7#) return if #6 is at least one

Question: Give the name of the student in the History department with the most credits.
Answer Steps: 1#) return students 2#) return #1 in History department 3#) return credits of #2 4#) return number of #3 for each #2 5#) return #2 where
        #4 is highest 6#) return name of #5

Question: when did lil wayne first start singing?
Answer Steps: 1#) return lil wayne 2#) return date that #1 start singing

Question: If there are no less than five dogs
Answer Steps: 1#) return dogs 2#) return number of #1 3#) return if #2 is at least five

Question: when was the last time the steelers won back to back super bowls
Answer Steps: 1#)
```

Figure 18: Example prompt for generating question decompositions for Break.

```
%geobase.pl
:- ensure_loaded(library('lists')).
:- ensure_loaded(library('ordsets')).
:- ensure_loaded(geobase).

country(countryid(usa)).

state(stateid(State)) :- state(State,_,_,_,_,_,_,_,_,_).

city(cityid(City,St)) :- city(_,St,City,_).

river(riverid(R)) :- river(R,_,_).

lake(lakeid(R)) :- lake(R,_,_).

mountain(mountainid(M)) :- mountain(_,_,M,_).

place(placeid(P)) :- highlow(_,_,P,_,_,_).
place(placeid(P)) :- highlow(_,_,_,_,P,_).

abbreviation(stateid(State), Ab) :- state(State,Ab,_,_,_,_,_,_,_,_).
abbreviation(Ab) :- abbreviation(_,Ab).

capital(stateid(State), cityid(Cap,St)) :- state(State,St,Cap,_,_,_,_,_,_,_).
capital(Cap) :- capital(_,Cap).

loc(X,countryid(usa)) :- city(X) ; state(X) ; river(X) ; place(X) ; lake(X); mountain(X).
loc(cityid(City,St), stateid(State)) :- city(State, St, City,_).
loc(cityid(City,St), stateid(State)) :- state(State,St,City,_,_,_,_,_,_,_).
loc(placeid(P), stateid(S)) :- highlow(S,_,P,_,_,_).
loc(placeid(P), stateid(S)) :- highlow(S,_,_,_,P,_).
loc(mountainid(P), stateid(S)) :- mountain(S,_,P,_).
loc(riverid(R), stateid(S)) :- river(R,_,States), member(S,States).
loc(lakeid(L),stateid(S)) :- lake(L,_,States), member(S,States).

traverse(riverid(R), stateid(S)) :- river(R,_,States), member(S,States).
traverse(riverid(R), countryid(usa)).

high_point(countryid(usa), placeid('mount mckinley')).
high_point(stateid(S), placeid(P)) :- highlow(S,_,P,_,_,_).

low_point(countryid(usa), placeid('death valley')).
low_point(stateid(S), placeid(P)) :- highlow(S,_,_,_,P,_).

area(stateid(X),Areal) :- state(X,_,_,_,Area,_,_,_,_,_), Areal is float(Area).
area(countryid(X),Areal) :- country(X,_,Area), Areal is float(Area).

major(cityid(C,S)) :- X = cityid(C,S), city(X), population(X,P), P > 150000.
major(riverid(R)) :- X = riverid(R), river(X), len(X,L), L > 750.
(omitted to save space)
%Translate the natural language description to prolog commands.

%Natural Language: what state is the biggest ?

%Natural Language: what is the population of montana ?

%Natural Language: what are the major cities in delaware ?

%Natural Language: what is the capital of michigan ?

%Natural Language: name the rivers in arkansas .

%Natural Language: what state has the largest urban population ?

%Natural Language: what is the longest river that flows through colorado ?

%Natural Language: what is the biggest state in continental us ?

%Natural Language: which states have points that are higher than the highest point in texas ?

%Natural Language: what is the longest river in the smallest state in the usa ?

%Natural Language:
```

Figure 19: Example prompt for generating questions for GeoQuery.

```
%geobase.pl
:- ensure_loaded(library('lists')).
:- ensure_loaded(library('ordsets')).
:- ensure_loaded(geobase).

country(countryid(usa)).

state(stateid(State)) :- state(State,_,_,_,_,_,_,_,_,_).

city(cityid(City,St)) :- city(_,St,City,_).

river(riverid(R)) :- river(R,_,_).

lake(lakeid(R)) :- lake(R,_,_).

mountain(mountainid(M)) :- mountain(_,_,M,_).

place(placeid(P)) :- highlow(_,_,P,_,_,_).
place(placeid(P)) :- highlow(_,_,_,_,P,_).

abbreviation(stateid(State), Ab) :- state(State,Ab,_,_,_,_,_,_,_,_).
abbreviation(Ab) :- abbreviation(_,Ab).

capital(stateid(State), cityid(Cap,St)) :- state(State,St,Cap,_,_,_,_,_,_,_).
capital(Cap) :- capital(_,Cap).

loc(X,countryid(usa)) :- city(X) ; state(X) ; river(X) ; place(X) ; lake(X); mountain(X).
loc(cityid(City,St), stateid(State)) :- city(State, St, City,_).
loc(cityid(City,St), stateid(State)) :- state(State,St,City,_,_,_,_,_,_,_).
loc(placeid(P), stateid(S)) :- highlow(S,_,P,_,_,_).
loc(placeid(P), stateid(S)) :- highlow(S,_,_,_,P,_).
loc(mountainid(P), stateid(S)) :- mountain(S,_,P,_).
loc(riverid(R), stateid(S)) :- river(R,_,States), member(S,States).
loc(lakeid(L),stateid(S)) :- lake(L,_,States), member(S,States).

traverse(riverid(R), stateid(S)) :- river(R,_,States), member(S,States).
traverse(riverid(R), countryid(usa)).

high_point(countryid(usa), placeid('mount mckinley')).
high_point(stateid(S), placeid(P)) :- highlow(S,_,P,_,_,_).

low_point(countryid(usa), placeid('death valley')).
low_point(stateid(S), placeid(P)) :- highlow(S,_,_,_,P,_).

area(stateid(X),Areal) :- state(X,_,_,_,Area,_,_,_,_,_), Areal is float(Area).
area(countryid(X),Areal) :- country(X,_,Area), Areal is float(Area).

major(cityid(C,S)) :- X = cityid(C,S), city(X), population(X,P), P > 150000.
major(riverid(R)) :- X = riverid(R), river(X), len(X,L), L > 750.
(omitted to save space)
%Translate the natural language description to prolog commands.

%Natural Language: what state is the biggest ?
answer(A,largest(A,state(A)))

%Natural Language: what is the population of montana ?
answer(A,(population(B,A),const(B,stateid(montana))))

%Natural Language: what are the major cities in delaware ?
answer(A,(major(A),city(A),loc(A,B),const(B,stateid(delaware))))

%Natural Language: what is the capital of michigan ?
answer(A,(capital(A),loc(A,B),const(B,stateid(michigan))))

%Natural Language: name the rivers in arkansas .
answer(A,(river(A),loc(A,B),const(B,stateid(arkansas))))

%Natural Language: what state has the largest urban population ?
answer(A,largest(B,(state(A),population(A,B))))

%Natural Language: what is the longest river that flows through colorado ?
answer(A,longest(A,(river(A),traverse(A,B),const(B,stateid(colorado)))))

%Natural Language: what is the biggest state in continental us ?
answer(A,largest(A,(state(A),loc(A,B),const(B,countryid(usa)))))

%Natural Language: which states have points that are higher than the highest point in texas ?
answer(A,(state(A),loc(B,A),higher(B,C),highest(C,(place(C),loc(C,D),const(D,stateid(texas))))))

%Natural Language: what is the longest river in the smallest state in the usa ?
answer(A,longest(A,(river(A),loc(A,B),smallest(B,(state(B),loc(B,C),const(C,countryid(usa)))))))

%Natural Language: count the states which have elevations lower than what alabama has .
answer(
```

Figure 20: Example prompt for generating prolog commands for GeoQuery.

```
%geobase.pl
:- ensure_loaded(library('lists')).
:- ensure_loaded(library('ordsets')).
:- ensure_loaded(geobase).
country(countryid(usa)).
state(stateid(State)) :- state(State,_,_,_,_,_,_,_,_).
city(cityid(City,St)) :- city(_,St,City,_).
river(riverid(R)) :- river(R,_,_).
…

%Translate the natural language description to prolog
commands.

%Natural Language: what state is the biggest?
answer(A,largest(A,state(A)))

%Natural Language: what is the population of montana?
answer(A,(population(B,A),const(B,stateid(montana))))

%Natural Language: count the states which have
elevations lower than what alabama has .
answer(
```

```
CREATE TABLE "border_info" ("state_name" text, "border"
text)
/*
3 example rows from table border_info:
SELECT * FROM border_info LIMIT 3;
state_name    border
  alabama tennessee
  alabama   georgia
  alabama   florida
*/
…

-- Using valid SQLite, answer the following questions for
the tables provided above.

-- Question: what state is the biggest ?
SELECT state.state_name FROM state WHERE
state.area=(SELECT max(state.area) FROM state);

-- Question: what is the population of montana ?
SELECT state.population FROM state WHERE
state.state_name='montana';

-- Question: count the states which have elevations lower
than what alabama has .
SELECT
```

(a) Prompt for GeoQuery.   (b) Prompt for GeoQuery-SQL.

Figure 21: Prompt comparison for GeoQuery and GeoQuery-SQL. Only two demonstrations and a part of symbolic knowledge are shown for simplicity.