# OpenReview forum: "Generating Data for Symbolic Language with Large Language Models"
_EMNLP/2023/Conference — EMNLP 2023 Main_

### Official Review · Reviewer_9jXq · 2023-08-02

**Soundness:** 4

**Excitement:**

3: Ambivalent: It has merits (e.g., it reports state-of-the-art results, the idea is nice), but there are key weaknesses (e.g., it describes incremental work), and it can significantly benefit from another round of revision. However, I won't object to accepting it if my co-reviewers champion it.

**Paper Topic And Main Contributions:**

This paper explores the problem of using LLMs as data generators for symbolic data generation with structured outputs, such as SQL or semantic parsing. The approach consists of a prompt that combines domain-specific symbolic knowledge and few-shot examples, along with a verification module to filter out low-quality examples, to generate a dataset of examples that can be used to fine-tune a task-specific model. In a number of experiments the authors demonstrate the efficacy of their approach. Highlights include the ability of the fine-tuned model to rival the performance of the full LLM and the SOTA in some instances; and the ability to match the performance of a model trained on 10x or 100x human-labeled data.

**Questions For The Authors:**

Instead of discarding the over-generated samples, have the authors considered using the 'bad' examples to contrastively train the downstream model? i.e. it should prefer generating a good example over a bad one?

**Reasons To Accept:**

The relevance of this paper to the state of the field is strong. LLMs are clearly powerful tools for a wide range of applications; however, they are not perfect and, in some cases, smaller, more custom models may still be preferred. Using LLMs as data generators for these situations are an obvious way to reduce the need for human effort.

The paper itself is well structured, and consists of a solid set of experiments that showcase the strengths and benefits of using LLMs as data generators for symbolic language.

**Reasons To Reject:**

There isn't a huge amount of novelty here. The main contribution, as far as I can tell, is the exploration of the capabilities of an off-the-shelf LLM for data generation. The greatest performance is gained from the inclusion of domain-specific knowledge and few-shot demonstrations to the prompt, neither of which are "engineered" by the authors; the natural language instructions, which are, appear to have the least impact on the outcome of the generator. The inclusion of agreement-based verification is interesting, but is also a fairly obvious way to validate outputs given the structured domains they are applied to.

Missing is a deeper look into how SymGen might be scaled to new domains. I assume you'd need: 1) Some domain-specific symbolic knowledge (how much? not necessarily a given for low-resource settings); 2) If applicable, a way to execute and validate generations (probably available, but not necessarily); 3) A set of few-shot demonstrations to draw from (possible to obtain, with the help of domain experts). A discussion about this would have been appreciated.

While most of the experiments are interesting and relevant, I find the inclusion of zero-shot generation results a bit strange here. I suppose this might satisfy general curiosity about the capabilities of the LLM in this setting, but it: 1) Clearly only works for symbolic domains that form a reasonable part of its training corpus; 2) This isn't a realistic setting since I assume getting 10 examples for a domain isn't that hard. A more interesting experiment may have been to explore the characteristics of the 10 examples and how that impacts final quality.

A more minor gripe I have is that the authors use the Appendix as though it were part of the paper and refer to results in the main body of the paper. The Appendix should only be used as supplementary material, and a reviewer should be able to arrive at a fair assessment of the paper without needing to actually refer to the Appendix. Otherwise what's the 8-page limit for anyway?

**Reproducibility:**

4: Could mostly reproduce the results, but there may be some variation because of sample variance or minor variations in their interpretation of the protocol or method.

**Reviewer Confidence:**

4: Quite sure. I tried to check the important points carefully. It's unlikely, though conceivable, that I missed something that should affect my ratings.

---

> ### Author Rebuttal · Authors · 2023-08-26
>
> We sincerely thank Reviewer 9jXq for the time you spent with our submission and your insightful suggestions and questions.
>
> **Missing is a deeper look into how SymGen might be scaled to new domains. I assume you'd need: 1) Some domain-specific symbolic knowledge (how much? not necessarily a given for low-resource settings); 2) If applicable, a way to execute and validate generations (probably available, but not necessarily); 3) A set of few-shot demonstrations to draw from (possible to obtain, with the help of domain experts). A discussion about this would have been appreciated.**
>
> Thanks for your great suggestion. The prompt engineering section investigates some ingredients, revealing that for languages like SQL, LLM can generate datasets of high quality without requiring domain-specific symbolic knowledge. However, for less common languages like TOP and Prolog, few-shot demonstrations and domain-specific symbolic knowledge are essential. We will add more discussion to our manuscript as suggested.
>
> **While most of the experiments are interesting and relevant, I find the inclusion of zero-shot generation results a bit strange here. I suppose this might satisfy general curiosity about the capabilities of the LLM in this setting, but it: 1) Clearly only works for symbolic domains that form a reasonable part of its training corpus; 2) This isn't a realistic setting since I assume getting 10 examples for a domain isn't that hard. A more interesting experiment may have been to explore the characteristics of the 10 examples and how that impacts final quality.**
>
> We completely agree with your point regarding the zero-shot setting and we sincerely appreciate your suggestion to explore the characteristics of the 10 examples in the few-shot setting. Currently, we randomly select the few-shot examples as a pilot study, and we believe characteristics such as diversity[1] should be important for the final quality of the dataset. We will update the paper once we have related results.
>
>
> **A more minor gripe I have is that the authors use the Appendix as though it were part of the paper and refer to results in the main body of the paper. The Appendix should only be used as supplementary material, and a reviewer should be able to arrive at a fair assessment of the paper without needing to actually refer to the Appendix. Otherwise what's the 8-page limit for anyway?**
>
> Thank you very much for your suggestions and we apologize for causing any complaints. As we are not allowed to change this manuscript in the rebuttal phase, we will rectify this problem in our future submissions.
>
>
> **Instead of discarding the over-generated samples, have the authors considered using the 'bad' examples to contrastively train the downstream model? i.e. it should prefer generating a good example over a bad one?**
>
> This is a great idea that we haven't tried before. However, we believe it should be helpful because the verifier can be viewed as a reward model, and methods in the alignment field (e.g., RLHF [2]) have demonstrated that training on top-preferred data (i.e., SFT) is always not sufficient. We think incorporating detailed signals from the verifier is a valuable future direction.
>
>
>
> References:
>
> [1] Hongjin, Su, et al. "Selective Annotation Makes Language Models Better Few-Shot Learners." The Eleventh International Conference on Learning Representations. 2022.
>
> [2] Ouyang, Long, et al. "Training language models to follow instructions with human feedback." Advances in Neural Information Processing Systems. 2022.

---

### Official Review · Reviewer_BFwA · 2023-08-02

**Typos Grammar Style And Presentation Improvements:** line 271, 280
**Soundness:** 4

**Excitement:**

4: Strong: This paper deepens the understanding of some phenomenon or lowers the barriers to an existing research direction.

**Paper Topic And Main Contributions:**

Large LMs have been used successfully to generate data via prompting for training smaller task-specific models, create knowledge graphs, etc. In this work, authors leverage this approach in the context of tasks mapping text to symbolic outputs (code, logical forms, etc) and demonstrate its effectiveness. Given a task dataset, they first generate questions via prompting the 175B Codex model and then prompt to generate responses to the generated questions. They show that effectiveness of generated data by finetuning T5-3B on it in addition to the original data which leads to a performance comparable to Codex on most tasks.
This approach is particularly effective in the setting where only a few (~10) human labelled samples are allowed. The authors then perform detailed ablations to show which components of the prompts are essential for the data quality, including analysis of generated data.

**Reasons To Accept:**

1. The setting is relevant and the method is effective - it is simple to execute and leads to large improvements in low data regime.
2. Experiments are comprehensive and convincing.
3. Performance comparable to strong baselines.
4. The generated data can be useful to others.

**Reasons To Reject:**

The method is not novel - but still the study is interesting so I dont have any major weaknesses to point out.


**Reproducibility:**

4: Could mostly reproduce the results, but there may be some variation because of sample variance or minor variations in their interpretation of the protocol or method.

**Reviewer Confidence:**

4: Quite sure. I tried to check the important points carefully. It's unlikely, though conceivable, that I missed something that should affect my ratings.

---

> ### Author Rebuttal · Authors · 2023-08-26
>
> We sincerely thank Reviewer BFwA for your review and are grateful for the time you spent on our submission. We are also glad you think our paper is effective and useful. Furthermore, many thanks for pointing out typos in our paper. We've revised them and will make every effort to improve the quality of our manuscript.

---

### Official Review · Reviewer_GgSV · 2023-08-05

**Soundness:** 4

**Excitement:**

4: Strong: This paper deepens the understanding of some phenomenon or lowers the barriers to an existing research direction.

**Missing References:**

 - [Shi+22]: Freda Shi, Daniel Fried, Marjan Ghazvininejad, Luke Zettlemoyer, and Sida I. Wang. 2022. Natural Language to Code Translation with Execution. In Proceedings of the 2022 Conference on Empirical Methods in Natural Language Processing, pages 3533–3546, Abu Dhabi, United Arab Emirates. Association for Computational Linguistics.

**Paper Topic And Main Contributions:**

This paper proposed to automatically create semantic parsing datasets by prompting LLMs, and such created datasets can be used to train a much smaller model (around 1% the parameter of LLMs) to get similar performance with prompting LLMs. This method requires much less human annotation while also drastically reducing the computational load when performing inference.

**Questions For The Authors:**

 - L138: How are the natural language queries generated? Solely by prompting an LLM to paraphrase? Please be clear in the main text.
 - L150: This section is not novel and has been proposed as _minimum Bayes risk decoding_ in [Shi+22].
 - Table 1: The similarity function for some code outputs are just _exact match_, which is very discrete for a tree-based formal language. Have you tried using tree edit distances, or S-Match [Cai+13]?
 - Table 2: How does "Codex + Verification" work? I assume that you oversample from Codex (e.g. by beam search), and then run the verification module. This method can also be used in T5 or T5+SymGen, which you have not applied verification to. Does verification help in these cases?

##### References
 - [Cai+13]: Shu Cai and Kevin Knight. 2013. Smatch: an Evaluation Metric for Semantic Feature Structures. In Proceedings of the 51st Annual Meeting of the Association for Computational Linguistics (Volume 2: Short Papers), pages 748–752, Sofia, Bulgaria. Association for Computational Linguistics.

**Reasons To Accept:**

 - Simple and intuitive method to create an artificial dataset. The method is highly reusable across domains.
 - Tested over 4 diverse semantic parsing tasks, therefore the conclusions are more convincing.


**Reasons To Reject:**

 - Lacking some citations (see below).
 - Details on the generation of the natural language questions are vague.

**Reproducibility:**

4: Could mostly reproduce the results, but there may be some variation because of sample variance or minor variations in their interpretation of the protocol or method.

**Reviewer Confidence:**

4: Quite sure. I tried to check the important points carefully. It's unlikely, though conceivable, that I missed something that should affect my ratings.

**Typos Grammar Style And Presentation Improvements:**

 - Figure 1 & 2: Consider using a monospace font for code output as it increases legibility.
 - L157, 160: $\mathrm{exec}=(\cdot)$, $\mathrm{sim}=(\cdot, \cdot)$: Remove the $=$ sign.
 - L269: "generate" => "generated".
 - L275: "github" => "GitHub".
 - L280: "sorely" => "solely".

---

> ### Author Rebuttal · Authors · 2023-08-26
>
> We sincerely thank Reviewer GgSV for your review and are grateful for the time you spent on our submission. Many thanks for pointing out typos in our paper.
>
> **L138: How are the natural language queries generated? Solely by prompting an LLM to paraphrase? Please be clear in the main text.**
>
> Thank you for pointing that out. The questions are generated by zero-shot or few-shot prompting. We have included some examples in the appendix, . For instance, we use the following instructions for NL2Bash task:
> ```
> Translate the natural language description to bash commands:
> Natural language:
> ```
> If few-shot examples are available, we can enrich the instruction as:
> ```
> Translate the natural language description to bash commands:
> Natural language: {EXAMPLE1}
> Natural language: {EXAMPLE2}
> ...
> Natural language:
> ```
> We will improve the relevant descriptions in the main text.
>
>
> **L150: This section is not novel and has been proposed as minimum Bayes risk decoding in [Shi+22].**
>
> Thank you for pointing out this great work that we missed. We will add them to the paper to make it more complete.
>
> **Table 1: The similarity function for some code outputs are just exact match, which is very discrete for a tree-based formal language. Have you tried using tree edit distances, or S-Match [Cai+13]?**
>
> We haven't tried it before, but it is indeed an interesting and great suggestion to enhance the verification and we'll explore it in future work.
>
>
> **Table 2: How does "Codex + Verification" work? I assume that you oversample from Codex (e.g. by beam search), and then run the verification module. This method can also be used in T5 or T5+SymGen, which you have not applied verification to. Does verification help in these cases?**
>
> Yes, your understanding of "Codex + Verification" is correct. However, SymGen also incorporates "verification" knowledge during the data generation phase. Since our main focus here is on how the data is generated,  we did not make further improvements to the inference of the smaller model (such as adding verification). However, based on the improvement of inference verification for Codex, we believe T5 and T5+SymGen should also benefit.

---

### Meta-Review · Area_Chair_Uh35 · 2023-09-25

**Recommendation:** 4

**Metareview:**

This paper proposes a method to automatically create semantic parsing datasets by prompting large language models (LLMs) and using the generated data to train smaller task-specific models. The authors demonstrate the effectiveness of this approach in the context of tasks that map text to symbolic outputs such as code or logical forms. They generate questions via prompting the 175B Codex model and then prompt to generate responses to the generated questions. The authors show that the generated data is effective by finetuning T5-3B on it in addition to the original data, which leads to a performance comparable to Codex on most tasks. This approach is particularly effective in settings where only a few (~10) human-labeled samples are allowed. The authors perform detailed ablations to show which components of the prompts are essential for data quality, including analysis of generated data.

Pros:
Simple and intuitive method to create an artificial dataset. The method is highly reusable across domains.
Tested over 4 diverse semantic parsing tasks, therefore the conclusions are more convincing.
The setting is relevant and the method is effective - it is simple to execute and leads to large improvements in low data regime.
Experiments are comprehensive and convincing and performance comparable to strong baselines.
The generated data can be useful to others.  The relevance of this paper to the state of the field is strong. Using LLMs as data generators for these situations are an obvious way to reduce the need for human effort.

Cons:
Lacking some citations.
Details on the generation of the natural language questions are vague.
The method is not novel - but still the study is interesting so I don’t have any major weaknesses to point out.
The paper is lacking novelty . The main contribution is the exploration of the capabilities of an off-the-shelf LLM for data generation.
The greatest performance is gained from the inclusion of domain-specific knowledge and few-shot demonstrations to the prompt.
Missing is a deeper look into how SymGen might be scaled to new domains.

---

### Decision · Program_Chairs · 2023-10-07

**Decision:**

Accept-Main

**Comment:**

This paper proposes a method to automatically create semantic parsing datasets by prompting large language models (LLMs) and using the generated data to train smaller task-specific models. The authors demonstrate the effectiveness of this approach in the context of tasks that map text to symbolic outputs such as code or logical forms. They generate questions via prompting the 175B Codex model and then prompt to generate responses to the generated questions. The authors show that the generated data is effective by finetuning T5-3B on it in addition to the original data, which leads to a performance comparable to Codex on most tasks. This approach is particularly effective in settings where only a few (~10) human-labeled samples are allowed. The authors perform detailed ablations to show which components of the prompts are essential for data quality, including analysis of generated data.

Pros:
Simple and intuitive method to create an artificial dataset. The method is highly reusable across domains.
Tested over 4 diverse semantic parsing tasks, therefore the conclusions are more convincing.
The setting is relevant and the method is effective - it is simple to execute and leads to large improvements in low data regime.
Experiments are comprehensive and convincing and performance comparable to strong baselines.
The generated data can be useful to others.  The relevance of this paper to the state of the field is strong. Using LLMs as data generators for these situations are an obvious way to reduce the need for human effort.

Cons:
Lacking some citations.
Details on the generation of the natural language questions are vague.
The method is not novel - but still the study is interesting so I don’t have any major weaknesses to point out.
The paper is lacking novelty . The main contribution is the exploration of the capabilities of an off-the-shelf LLM for data generation.
The greatest performance is gained from the inclusion of domain-specific knowledge and few-shot demonstrations to the prompt.
Missing is a deeper look into how SymGen might be scaled to new domains.